# Identification of circRNA-lncRNA-miRNA-mRNA Competitive Endogenous RNA Network as Novel Prognostic Markers for Acute Myeloid Leukemia

**DOI:** 10.3390/genes11080868

**Published:** 2020-07-31

**Authors:** Yaqi Cheng, Yaru Su, Shoubi Wang, Yurun Liu, Lin Jin, Qi Wan, Ying Liu, Chaoyang Li, Xuan Sang, Liu Yang, Chang Liu, Zhichong Wang

**Affiliations:** State Key Laboratory of Ophthalmology, Zhongshan Ophthalmic Center, Sun Yat-sen University, Guangzhou 510060, China; chengyq7@mail2.sysu.edu.cn (Y.C.); xha_000@163.com (Y.S.); wangshb6@mail2.sysu.edu.cn (S.W.); lyr9506@163.com (Y.L.); jinlin3@mail2.sysu.edu.cn (L.J.); wanq9@mail2.sysu.edu.cn (Q.W.); liuying@gzzoc.com (Y.L.); lichaoyang@gzzoc.com (C.L.); sangx@mail2.sysu.edu.cn (X.S.); yangliu28@mail2.sysu.edu.cn (L.Y.); liuchang@gzzoc.com (C.L.)

**Keywords:** acute myeloid leukemia, ceRNA network, prognosis

## Abstract

Background: Acute myeloid leukemia (AML) is one of the most common malignant and aggressive hematologic tumors, and its pathogenesis is associated with abnormal post-transcriptional regulation. Unbalanced competitive endogenous RNA (ceRNA) promotes tumorigenesis and progression, and greatly contributes to tumor risk classification and prognosis. However, the comprehensive analysis of the circular RNA (circRNA)-long non-coding RNA (lncRNA)-miRNA-mRNA ceRNA network in the prognosis of AML is still rarely reported. Method: We obtained transcriptome data of AML and normal samples from The Cancer Genome Atlas (TCGA), Genotype-tissue Expression (GTEx), and Gene Expression Omnibus (GEO) databases, and identified differentially expressed (DE) mRNAs, lncRNAs, and circRNAs. Then, the targeting relationships among lncRNA-miRNA, circRNA-miRNA, and miRNA-mRNA were predicted, and the survival related hub mRNAs were further screened by univariate and multivariate Cox proportional hazard regression. Finally, the AML prognostic circRNA-lncRNA-miRNA-mRNA ceRNA regulatory network was established. Results: We identified prognostic 6 hub mRNAs (TM6SF1, ZMAT1, MANSC1, PYCARD, SLC38A1, and LRRC4) through Cox regression model, and divided the AML samples into high and low risk groups according to the risk score obtained by multivariate Cox regression. Survival analysis verified that the survival rate of the high-risk group was significantly reduced (*p* < 0.0001). The prognostic ceRNA network of 6 circRNAs, 32 lncRNAs, 8 miRNAs, and 6 mRNAs was established according to the targeting relationship between 6 hub mRNAs and other RNAs. Conclusion: In this study, ceRNA network jointly participated by circRNAs and lncRNAs was established for the first time. It comprehensively elucidated the post-transcriptional regulatory mechanism of AML, and identified novel AML prognostic biomarkers, which has important guiding significance for the clinical diagnosis, treatment, and further scientific research of AML.

## 1. Introduction

Acute myeloid leukemia (AML) is a hematologic malignancy tumor characterized by abnormal proliferation and differentiation and maturation disorders of hematopoietic progenitor cells. It is the most common type of acute leukemia in adults [1]. The American Cancer Society predicts that by 2020 there will be an estimated 21,040 new AML patients in the United States, and 11,180 people will die from this disease [2]. AML is prone to enter the recurrence and refractory stage characterized by extramedullary malignant infiltration, involving multiple organs such as skin, lymph nodes, liver, spleen, and even central nervous system, indicating a poor prognosis [3]. The tumorigenesis of AML is associated with complex and variable gene mutations, making it of great heterogeneity in molecular characteristics, pathogenesis, and clinical manifestations. Although traditional intensive induction chemotherapy and post-remission treatment have delayed the disease progression to some extent, most patients still relapse or even die after achieving complete remission [4]. In recent years, precise cell therapy and monoclonal antibody therapy which target a variety of AML molecular markers have been on the rise [5]. However, it is still unable to fully cope with the complex types of mutations, suggesting that more researches on the pathogenesis of AML and exploration of the therapeutic targets have important clinical application value and transformation prospects.

With the development of the post-transcriptional regulatory mechanism associated research, the function of non-coding RNAs (ncRNAs) and the interaction between endogenous RNAs have been discovered: circular RNA (circRNA) is a newly discovered type of ncRNA, which composes of hundreds to thousands of nucleotides (nt) and is a closed circular single-strand. Its expression is tissue specific and has a highly sequence conservation between species. The main functions of circRNA are to bind through miRNA response element (MRE) and absorb RNA-binding protein. Meanwhile, many studies have also found that some circRNA can be translated [6,7]; long non-coding RNA (lncRNA) is defined as linear non-coding RNA with a length greater than 200 nt. It is widely involved in nuclear chromatin structure, gene transcription, and post-transcriptional regulation, and can act as a sponge to adsorb miRNA [8]. As non-coding lncRNAs, circRNAs, and protein-coding mRNAs all have MRE regions that can bind microRNAs (miRNAs), they compete for limited miRNAs and form a competitive endogenous RNA (ceRNA) regulatory network. When mRNA competes to bind miRNAs, its stability decreases, the translation process is blocked, and gene expression is affected, in which way a variety of ncRNAs participate in the regulation of mRNA coding function. The ceRNA network plays a role in multiple physiological and pathological processes [9]. It has been found that the abnormal ceRNA network regulates the occurrence and development of various cancers such as breast cancer, colon cancer, liver cancer, prostate cancer, bladder cancer, lung cancer, stomach cancer, and hematologic tumor [10].

The role of ceRNA networks in AML is also of concern. circRNA-DLEU2, which is highly expressed in AML, sponges miRNA-496 as a ceRNA, which promotes the expression of PRKACB gene, thereby promoting AML cell proliferation and inhibiting their apoptosis, and has been proved to promote tumor formation in vivo [11]. circRNA_100290 competes to bind miRNA-293, then upregulates the expression RAS signaling pathway member Rab10 gene, accelerating AML progression [12]. In pediatric AML, circRNA-0004136 sponges leukemia associated miRNA-142 and promotes tumor cells proliferation [13]. The highly expressed lncRNA RPPH1 in AML can act as ceRNA to adsorb miRNA-330-5P, and promote the growth, invasion, and migration of tumor cells in vivo and in vitro [14]. lncRNA-00662 sponges miRNA-340-5P to upregulate ROCK1 gene expression and promote malignant proliferation of AML cells [15].

With the rapid development of high-throughput sequencing and large sample analysis, bioinformatics analysis is widely used in oncology research and prognosis prediction. Analyzing public databases to construct a more comprehensive ceRNA regulatory network and to mine more accurate prognostic markers has been widely used. In recent years, there have been few studies to analyze the prognostic lncRNA-miRNA-mRNA [16] and circRNA-miRNA-mRNA [17] ceRNA network in AML. However, there is still no research to include circRNA and lncRNA simultaneously into the ceRNA network.

In our study, TCGA [18], Genotype-tissue Expression (GTEx) [19], and Gene Expression Omnibus (GEO) [20] databases were employed to obtain and identify DEmRNAs, DElncRNAs, and DEcircRNAs between AML and normal tissues, which ensures accuracy and repeatability of the analysis results. miRNAs targeted by lncRNAs and circRNAs were predicted separately and were taken intersections, while multiple databases were used to predict the targeted relationship between the common miRNAs and mRNAs. The survival prediction model was established to screen the hub mRNAs with high prognostic value, and the hub prognostic ceRNA regulatory network of 6 circRNAs, 32 lncRNAs, 8 miRNAs and 6 mRNAs, was finally constructed.

## 2. Materials and Methods

### 2.1. Transcriptome Data and Clinical Information

The RNA sequencing data of 151 AML patients’ bone marrow (BM) samples and 407 normal peripheral blood samples were respectively downloaded from TCGA database (https://portal.gdc.cancer.gov/) and GTEx database. The lncRNAs were annotated using Genecode (https://www.gencodegenes.org/) to distinguish them from mRNAs. The circRNA microarray GSE94591 (GPL19978 Agilent-069978 Arraystar Human CircRNA microarray V1) containing 3 high risk AML, 3 low risk AML, and 4 normal control samples’ BM monocytes circRNA expression data, and mRNA microarray GSE79605 (GPL6480 Agilent-014850 Whole Human Genome Microarray 4 × 44K G4112F) containing 2 AML and 2 normal control samples’ BM monocytes mRNA expression data were both downloaded from GEO database. The clinical information of 151 AML samples was obtained from TCGA database.

### 2.2. Identification of DERNAs

The transcriptome data were normalized using the “normalizeBetweenArrays” function of “limma” R package. The DEmRNAs, DElncRNAs, and DEcircRNAs between AML and normal samples were identified with same R package with |log2FC| > 1.5 with adjusted *p*-value < 0.05 as the threshold for statistical significance. The “pheatmap” and “ggplot2” R packages were used to draw heatmap and volcanos of the DERNAs. The Venn diagram was employed to take intersection of DERNAs.

### 2.3. Gene Function Annotation

Gene ontology (GO) analysis was conducted on the primary screened mRNAs to assess enrichment for biological processes (BP), cellular component (CC), and molecular function (MF) annotations. Kyoto Encyclopedia of Genes and Genomes (KEGG) analysis was also performed to annotate the signaling pathways associated with these DEmRNAs. Both GO and KEGG enrichment were performed using the “clusterProfiler” R package, with adjusted *p* < 0.05 as the threshold for statistical significance. The “ggplot2” R package was used to visualize the GO and KEGG analysis results.

### 2.4. Prediction of Targeting Relationship

We selected the top three down-regulated and up-regulated circRNAs as the hub components among the DEcircRNAs referring to recent studies [21,22], and obtained the annotation of the 6 DEcircRNA from the circBase database [23]. We predicted the miRNA (circ-pre-miRNAs) targeted by these circRNAs, and obtained the structure, miRNA response element (MRE), and RNA-binding protein (RBP) information from Cancer-specific circRNA (CSCD) Database [24]. The miRcode database was employed to predict the DElncRNAs targeting miRNAs (lnc-pre-miRNAs) [25]. The overlap (intersection) between DEcircRNA predicted miRNAs and DElncRNA predicted miRNA (pre-miRNAs) was then evaluated, mRNAs (pre-mRNAs) targeted by common miRNAs were predicted by Targetscan [26] and miRTarBase [27] database. Cytoscape v3.7.2 was used to construct the initial TME-related ceRNA network [28].

### 2.5. Survival Prediction Model Identifies Hub mRNAs

Cox proportional hazard regression takes survival outcome and survival time as dependent variables, and is an effective method to simultaneously analyze the impact of various factors on survival. Univariate Cox proportional hazards regression analysis was employed to identify the relationship between the mRNAs expression and overall survival (OS) of patients, the mRNAs with *p* < 0.05 will be performed multivariate Cox proportional hazards regression to identify hub prognostic mRNAs [29]. Subsequently, using the median of the risk scores obtained by multivariate Cox regression as the boundary, the AML patients were divided into high and low risk groups. The “survival” R package was employed to perform survival analysis and draw survival curves of the two groups. The receiver operating characteristic (ROC) curve was established by “survival ROC” R package.

### 2.6. ceRNA Network Establishment

A circRNA–lncRNA–miRNA–mRNA ceRNA network with hub prognostic mRNAs as the core was constructed based on the targeting relationships. This ceRNA network revealed the competitive binding of miRNAs by circRNAs, lncRNAs, and mRNAs, thereby established a complex post-transcriptional regulatory network in AML.

### 2.7. Statistical Analysis

All statistical analyses were performed using R software (v.3.6.1) and the aforementioned packages.

## 3. Results

### 3.1. Identification of DEmRNAs between AML and Normal Samples

The expression profiles of mRNAs in 151 bone marrow samples with AML and 407 normal whole blood samples were explored. The clinicopathological and molecular characteristics of AML patients were shown in Table 1. After normalized the expression profiles using the “normalizeBetweenArrays” function of “limma” R package, We applied |log_2_FC| > 1.5, adjusted *p*-value < 0.05 as the standard to screened out 3078 DEmRNAs between TCGA AML samples and GTEx normal control samples (Figure 1a), and obtained 1774 DEmRNAs from GEO microarray GSE79605 with the same threshold (Figure 1b). To preliminarily screen for mRNAs with close correlation with AML, DEmRNAs from different databases were intersected and 356 common mRNAs were obtained, among which 95 were up-regulated and 261 were down-regulated. GO enrichment analysis of the common mRNAs showed that they were mainly enriched in biological processes such as neutrophil degranulation and activation (GO: 0043312, GO: 0002283, GO: 0042119, GO: 0002446), and cytokine production (GO: 0001819) (Figure 1c, Table 2). KEGG signaling pathway enrichment showed that these common genes were significantly participated in hematopoietic cell lineage (hsa04640), antigen presentation (hsa04612), lysosomes, phagosomes, and other pathways associated with blood system, infection, and immunity (Figure 1d, Table 3).

### 3.2. Identification of DelncRNAs and DEcircRNAs between AML and Normal Samples

We applied |log_2_FC| > 1.5, adjusted *p*-value < 0.05 as the standard to analyze lncRNA data from TCGA and GTEx databases and circRNA data from GEO microassay GSE94591, and screened out 89 DElncRNAs (Figure 2a,b) and 172 DEcircRNAs (Figure 3a).

### 3.3. Prediction and Identification miRNAs Targeted by Both lncRNA and circRNA

The miRcode database was employed to predict the targeting miRNA of 89 DElncRNAs. Since lncRNAs and miRNAs are not one-to-one corresponded, we obtained the interaction relationship between 58 lncRNAs and 85 miRNAs (Figure 2c).

We selected the top three (with biggest |log_2_FC|) up-regulated (hsa_circ_0001247, hsa_circ_0012152, and hsa_circ_0003602) and down-regulated (hsa_circ_0005571, hsa_circ_0007609, and hsa_circ_0074371) circRNAs as the hub components among the DEcircRNAs referring to recent studies. Genes where the 6 circRNAs derived from were located on chromosomes 22, 1, 3, 19, 20, and 5 (Figure 3b). Using the chromosomal location of circRNA-derived genes, we obtained the structural information and targeting miRNAs of 6 circRNAs in CSCD database, and finally established the targeting relationship between 6 circRNAs and 323 miRNAs (Figure 3e).

The circ-pre-miRNAs and linc-pre-miRNAs were intersected to obtain 49 common miRNAs, which were used as the common competitive binding targets to initially reveal the relationship between circRNAs and lncRNAs (Figure 3c–d).

### 3.4. Prediction of miRNA-mRNA Targeting Relationship and Screening for mRNAs Highly Related to AML

MRNAs targeted by 49 pre-miRNAs was predicted by Targetscan and miRTarBase databases. There were 3167 mRNAs predicted by both two databases. In order to further improve the correlation between ceRNA network and AML, the pre-mRNAs and the common DEmRNAs from TCGA, GTEx, and GEO databases were further intersected to obtain 69 hub mRNAs, among which 31 mRNAs were up-regulated and 38 mRNAs were down-regulated (Figure 4a).

### 3.5. Hub AML Prognostic mRNAs Screened by Cox Regression

Univariate Cox proportional hazard regression analysis was employed to verify the correlations between expression levels of these 69 common mRNAs and overall survival (OS), and 14 prognostic mRNAs were identified according to threshold with *p* < 0.05. These 14 mRNAs were further analyzed by multivariate Cox proportional hazard regression analysis, and 6 hub mRNAs were selected to predict the prognosis of AML: TM6SF1, ZMAT1, MANSC1, PYCARD, SLC38A1, and LRRC4 (Table 4). The expression of ZMAT1T and SLC38A1 increased in AML samples, while the expression of TM6SF1, MANSC1, PYCARD, and LRRC4 decreased in AML samples (Figure 4f), and the correlation between the 6 mRNAs was verified (Figure 4e). The risk score formula for OS was calculated as follows: risk score = −0.5585 × (expression value of TM6SF1) + −0.7650 × (expression value of ZMAT1) + 0.5320 × (expression value of MANSC1) + 1.2461 × (expression value of PYCARD) + 0.6975 × (expression value of SLC38A1) + −0.6860 × (expression value of LRRC4). Kaplan–Meier survival analysis of high and low risk groups showed that the survival rate of the high-risk group was significantly lower than that of the low-risk group (*p* < 0.00001, Figure 4b). To test the prediction accuracy of the model, we constructed ROC curves based on the risk scores obtained from the multivariate Cox regression. The area under the 3-year survival curve (AUC) was 0.784, indicating high accuracy and specificity (Figure 4c). In addition, we verified the expression of 6 hub mRNAs in the high- and low-risk groups, which were consistent with their expression trends between AML and normal samples (Figure 4d).

Considering that AML is a heterogenous disease and the outcomes of different French–American–British (FAB) classifications vary greatly, we analyzed the differences in the expression levels of 6 hub mRNAs between M0-M7 subtypes (Figure 5a). The result showed that the expressions of the 6 mRNAs in different subtypes were significantly different, and the expression distribution trend are obvious differences among these mRNAs. TM6SF has the highest expression value in M7 subtype and the lowest expression value in M6 (*p* = 0.00013), ZMAT1 has the highest expression value in M0 subtype and the lowest expression value in M7 subtype (*p* < 0.0001), MANSC1 was mostly expressed in M6 subtype and the minimally expressed in M3 subtype (*p* = 0.00045), PYCARD has the highest expression value in M5 subtype, and the lowest expression value in M7 subtype (*p* < 0.0001), SLC38A1 has the highest expression value in M0 subtype and the lowest expression value in M3 subtype (*p* < 0.0001), LRRC4 has the highest expression value in M4 subtype and he lowest expression value in M6 subtype (*p* = 0.00315).

At the same time, we also analyzed the differences in the expression levels of 6 core mRNAs among the three cytogenetic risk groups (Figure 5b). It was also found that the distribution of 6 mRNAs in different cytogenetic risk groups was significantly different. TM6SF (*p* < 0.0001) and LRRC4 (*p* = 0.0449) had the highest expression value in the samples with the favorable karyotype, ZMAT1 (*p* = 0.0355), MANSC1 (*p* = 0.0389), and PYCARD (*p* = 0.0017) has the highest expression value in the intermediate karyotype samples, while SLC38A1 (*p* = 0.0047) has the highest expression in the poor outcome group samples. These results were consistent with the expression trend of hub mRNAs in AML compared to normal controls, and were also consistent with the results of multivariable Cox, which the expression of TM6SF, LRRC4, MANSC1, and PYCARD decreased in AML, and the increased expression of ZMAT1 and SLC38A1 in AML samples. In order to establish a more refined scoring system to evaluate the impact of hub mRNAs on prognosis, we further constructed a nomogram of 6 mRNAs based on the multivariable Cox result (Figure 5c).

### 3.6. Establishment of AML Prognostic circRNA-lncRNA-miRNA-mRNA ceRNA Network

The 6 hub prognostic mRNAs were predicted to interact with 8 miRNAs, hsa-mir-590-3p, hsa-mir-27a-5p, hsa-mir-383-3p, hsa-mir-34c-3p, hsa-mir-17-3p, hsa-mir-215-3p, hsa-mir-506-5p, and hsa-mir-182-5p, which in turn were predicted to interact with 32 lncRNAs and 6 circRNAs. Based on the above relationship, we constructed a hub prognostic ceRNA network consisting of 6 mRNAs, 8 miRNAs, 32 lncRNAs, and 6 circRNAs (Figure 6, Table 5). At the same time, Figure 7 is drawn to show the molecular mechanism of the AML hub prognostic ceRNA network more intuitively.

The analysis flow of this study is shown in Figure 8.

## 4. Discussion

ceRNA regulatory network theory points out that various RNAs share common MRE, which enables them competitively bind miRNA to realize mutual regulation. ceRNA network has been proved to be ubiquitous in post-transcriptional regulation of gene expression, and its role in a variety of diseases, especially tumors, has attracted wide attention. Cumulative studies are exploring and discussing the role of ceRNA in tumor prognosis. The pathogenesis of AML is complex, heterogeneous, and is still not fully elucidated. At the same time, in the face of high recurrence rate and mortality, there is still a lack of reliable prognostic targets. In order to better understand the abnormal gene expression regulation of AML and look for possible prognostic markers, we analyzed multiple databases to obtain endogenous RNA with abnormal expression in AML, and established a hub prognostic ceRNA regulatory network through multiple screening and verification.

The ceRNA regulatory network plays an important role in tumorigenesis and development, and is used to the study of pathogenesis, diagnosis, treatment, and the prognosis prediction of most tumors, including AML. This study comprehensively analyzed the AML transcriptome expression data in the TCGA, GTEx, and GEO databases, established a ceRNA network, and confirmed that it is a reliable prognostic target for AML. In recent years, the role of ceRNA in diseases has become increasingly prominent. The study of Song et al. confirmed that in gastric cancer, lncRNA-KRTAP5-AS1 and lncRNA-TUBB2A promote the high expression of Claudin-4, which is related to poor prognosis of gastric cancer through competitive binding miRNA-596 and miRNA-3620-3P, promoting the proliferation, invasion and metastasis of gastric cancer cells [30]. Meng et al. found that in glioma, up-regulation of circSCAF11 expression can promote glioma neovascularization through the miRNA-421-SP1-VEGF axis, and is closely related to the poor clinical prognosis of the disease [31]. Chen et al. proved that during intrahepatic vascular metastasis of hepatocellular carcinoma, the expression of lncRNA-CDKN2B-AS was increased, and the expression of metastasis-related gene ARHGAP18 was upregulated through the competitive binding of miRNA-153-5p by lncRNA-CDKN2B-AS [32]. In AML, some researchers used the Cox proportional hazards regression model to screen out 8 mRNAs with prediction value, and constructed a ceRNA network consisting of lncRNAs, miRNAs, and mRNAs according to the mutual targeting relationship [16]. In another study, the regulatory network of 17 circRNAs and 7 mRNAs was screened from RNA sequencing to explore the biomarkers targeting AML extramedullary infiltration [17]. In addition, ceRNA networks are also used to predict prognosis of lung cancer [33], bladder cancer [34], endometrial cancer [35], rectal cancer [36], and other cancers. However, previous studies on AML ceRNA regulatory networks have only explored the role of either lncRNA or circRNA, and there has no comprehensive screening of circRNA, lncRNA, miRNA, and mRNA simultaneously to construct AML prognostic ceRNA networks.

The data analyzed in this study comes from multiple platforms, which not only guarantees a sufficient and rich sample size, but also eliminates the possible bias of a single platform analysis. Meanwhile, the reliability and high AML correlation of the screened hub mRNAs were further improved by taking intersection of target prediction and differential expression.

The prognostic signature screening model constructed with univariate Cox, multivariate Cox, Kaplan–Meier survival analysis, and nomogram as the hub in this study fully ensures the credibility and accuracy of the results. In this study, 6 prognostic mRNAs (TM6SF1, ZMAT1, MANSC1, PYCARD, SLC38A1, and LRRC4) were used as the core foundation to construct a more comprehensive ceRNA regulatory network related to circRNA, lncRNA, miRNA, and mRNA as the target biomarker for AML prognosis prediction.

In this study, 6 hub mRNAs in the risk signature divided AML patients into high-risk groups and low-risk groups. Kaplan–Meier analysis confirmed that the high-risk group had a significantly worse prognosis. The ceRNA screened in this study is of great significance in AML and many tumors. PYCARD, also known as ASC, a classic key protein that promotes apoptosis [37,38]. The decreased expression of PYCARD causes cells to enter a state of anti-apoptosis and abnormal proliferation, which is closely related to tumorigenesis. Mhyre AJ et al. found that normal BM stromal cells can up-regulate the expression of PYCARD in myelodysplastic syndrome tumor cells to enhance their sensitivity to pro-apoptotic Tumor necrosis factor (TNFα) molecules [39]. Transfection of miRNA-143 in acute T cell leukemia cell line Jurket can up-regulate PYCARD expression, revealing the potential role as ceRNA of PYCARD [40]. In addition, the silencing of PYCARD after methylation promotes the development of breast cancer, colon cancer, and other tumors.

SLC38A1 is a member of the amino acid transport family, and it has been confirmed that the survival time of patients with increased SLC38A1 expression in AML is significantly shorter than that of patients with low expression [41]. Meanwhile, the high expression of SLC38A1 is closely related to the occurrence, recurrence, and metastasis of solid tumors such as gastric cancer [42], colon cancer [43], and cervical cancer, indicating poor prognosis.

TM6SF1 is a six-time transmembrane protein located on the lysosomal membrane and plays an important role in intracellular and extracellular protein transport. Studies have shown that the methylation level of TM6SF1 gene in breast cancer patients’ cancer tissues and breast milk is significantly higher than that in healthy people [44,45]. Hypermethylation of TM6SF1 gene also appears in HBV-related liver cancer [46]. Hypermethylation corresponds to low gene expression, suggesting that the above researches are consistent with the results of our analysis that expression of TM6SF1 is decreased in AML.

LRRC4 is a classic glioma suppressor gene. Many studies have shown that miRNA-38147, miRNA-182, and miR-381 48 can bind to LRRC4 and inhibit its expression, promoting the progression of glioma [47,48], In addition, the decreased expression of LRRC4 was associated with the occurrence and development of hematological non-Hodgkin’s lymphoma [49] and solid tumors such as meningioma, pituitary tumor [50], and nasopharyngeal carcinoma [51]. MANSC1 has been reported to be expressed in BM samples of most patients with acute and chronic leukemia and other hematological tumors [52]. Nucleotide mutation of MANSC1 protein may be associated with prostate cancer [53].

When analyzed the expression differences of the 6 hub prognostic mRNAs between different AML subtypes and karyotype risk groups, we found that different mRNAs have different expression tropisms. For example, TM6SF has the highest expression value in M7 subtype and the lowest expression value in M6, while ZMAT1 has the highest expression value in M0 subtype and the lowest expression value in M7 subtype. This result suggests that although this signature can reflect the prognosis of AML as a whole, there may be differences in detail in different components involved in the pathophysiological processes in AML, and they may be more representative of the prognosis of a certain subtype.

Similarly, the screened 8 hub miRNAs and 24 of the 32 hub lncRNAs in the ceRNA network, including PCBP1-AS1, ITCH-IT1, GAS5, and MALAT1 have been confirmed to be involved in post-transcriptional regulation of breast cancer, liver cancer, colon cancer, bladder cancer, lung cancer, and other tumors. They have become new targets for tumor diagnosis and prognosis prediction. [54,55,56,57,58]. Among the hub circRNAs, hsa_circ_0007609, also known as circ-DNAJC5 according to its source gene, is considered to be related to the prognosis of multiple myeloma in the study of Zhou et al. [59]. Furthermore, by querying the CSCD database, we found that the six hub circRNAs in this study are all related to the hematopoietic system, leukemia, or other tumors. However, expression of hsa_circ_0074371 is up-regulated in gastric cancer, and inhibiting its expression can promote gastric cancer cell apoptosis [60], which is inconsistent with our research result that its expression in AML is reduced, suggesting that the same circRNA may have different roles in different tumors, we suspect this may be related to the differences of downstream miRNA, mRNA, and signaling pathway. The expression of lncRNA-LINC00173 is reduced in non-small cell lung cancer, causing miRNA-182 to aggregate and inhibit the Advanced Glycosylation End-Product Specific Receptor (AGER) / nuclear factor kappa-B (NF-κB) pathway, promoting tumor growth and metastasis [61]. lncRNA-CRNDE can competitively bind miRNA-181, up-regulate the expression of Wnt pathway, promote the growth and drug resistance of colon cancer [62]. lncRNA-SNHG6 competitively binds miRNA-125 to activate snail pathway in bladder cancer [63]. In summary, a large number of previous studies have confirmed that the ceRNA network we constructed indeed regulates the tumor progression, plays an important role in prognosis prediction.

Although we tried our best to improve the representativeness of AML-related ceRNAs and the reliability of the prognosis through multiple levels of verification, there are still some deficiencies: According to the reference, we use the whole peripheral blood RNA sequencing data of GTEx database as a normal control [16], which is inconsistent with the BM-derived samples in AML group, suggesting that more normal bone marrow sample sequencing data will be urgently needed in future research. Due to the limitation that the expression data of various RNAs are obtained from different platforms and different samples, we currently failed to merge them together for co-expression analysis or other analysis for calculation of ceRNAs’ correlation and connectivity. The source of the sample should be considered with the purpose to identify the scores of ceRNA interaction. In addition, the role of ZMAT1 gene and some hub lncRNAs, circRNAs is still lack of reports in AML. What is more, we verified the targeting relationship through multiple databases, we would like to fully verify it in future experiments and explore its feasibility for clinical application.

## 5. Conclusions

In conclusion, this study established an AML-related ceRNA regulatory network of circRNA-lncRNA-miRNA-mRNA for the first time, proving that it is significantly correlated with the prognosis of AML. We also more comprehensively described the post-transcriptional regulatory mechanism of AML, provided a new direction for understanding the pathogenesis, and found more effective targets for diagnosis, treatment, and prognosis prediction.

## Figures and Tables

**Figure 1 genes-11-00868-f001:**
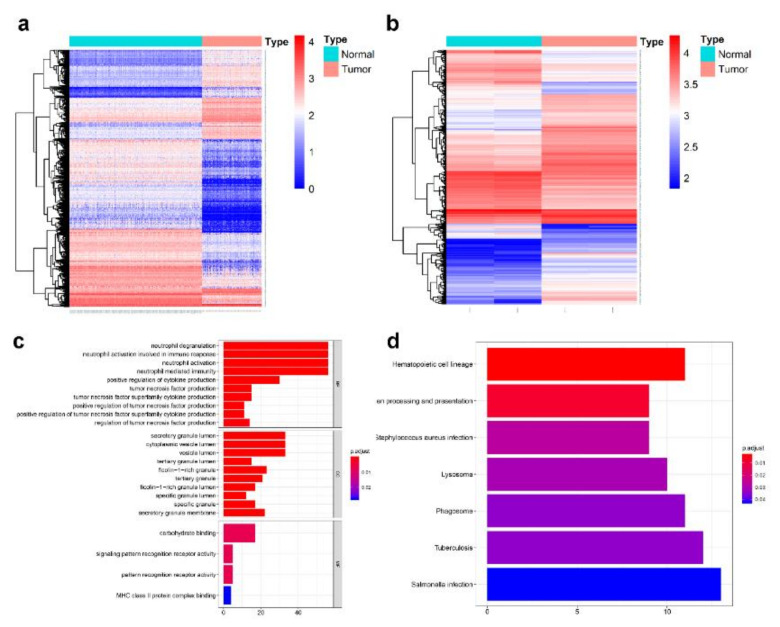
Identification and functional analysis of differentially expressed mRNAs (DEmRNAs). (**a**,**b**) Heatmap of acute myeloid leukemia (AML) DEmRNAs analyzed from The Cancer Genome Atlas (TCGA), Genotype-tissue Expression (GTEx), and Gene Expression Omnibus (GEO) databases. Red color represents increased expression, blue color represents decreased expression. The darker the color is, the greater the difference of mRNAs expression. (**c**) Bar diagram the GO function annotation result of common DEmRNAs. (**d**) Bar diagram of the Kyoto Encyclopedia of Genes and Genomes (KEGG) function enrichment of common DEmRNAs.

**Figure 2 genes-11-00868-f002:**
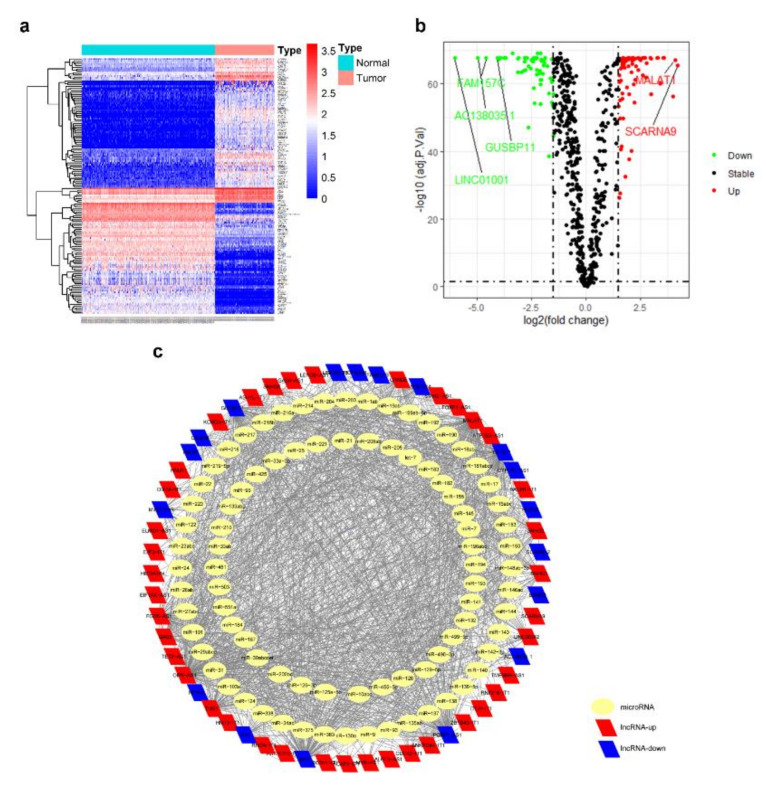
Identification of DElncRNAs and prediction of targeted miRNAs. (**a**) Heatmap of AML DElncRNAs from TCGA and GTEx databases. (**b**) Volcano plot of AML DElncRNAs from TCGA and GTEx databases. (**c**) Prediction of DElncRNAs targeting miRNAs. The interaction network was constructed consisting of 58 lncRNAs and 85 miRNAs.

**Figure 3 genes-11-00868-f003:**
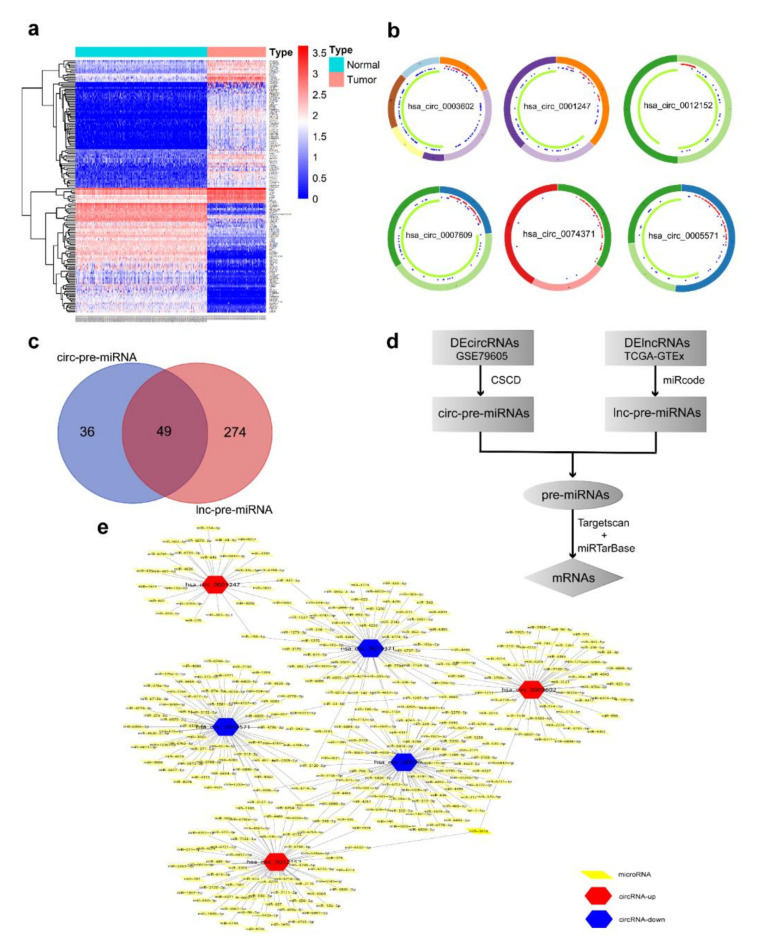
Identification of DEcircRNAs and prediction of targeted miRNAs. (**a**) Heatmap of AML DEcircRNAs from GEO databases. (**b**) Structure annotation of circRNAs. (**c**) Venn diagram of the intersection of pre-miRNAs. A total of 49 miRNAs were commonly targeted by both lncRNAs and circRNAs. (**d**) Flow chart of common pre-miRNAs prediction. (**e**) Prediction of circRNA-targeted miRNAs. The interaction network was constructed with 6 circRNAs and 323 miRNAs.

**Figure 4 genes-11-00868-f004:**
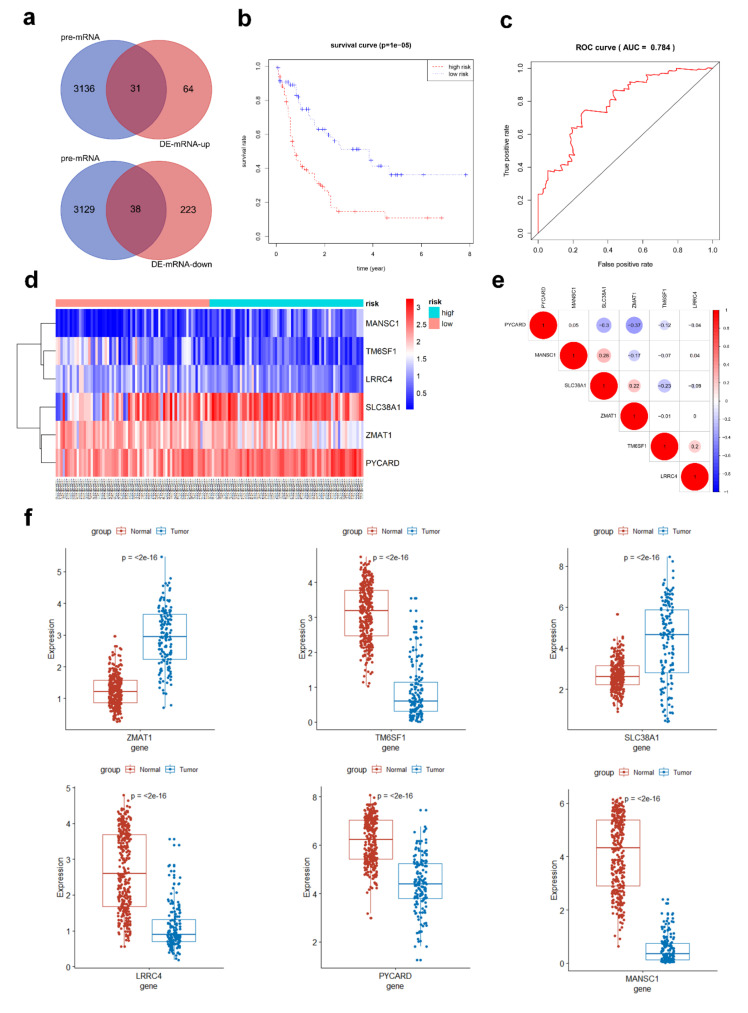
Identification of hub prognostic mRNAs through Cox regression and survival analysis. (**a**) The pre-mRNAs and common DEmRNAs were intersected, and 69 hub mRNAs were preliminary obtained. (**b**) Survival curve of patients in different risk groups. The patients were divided into high and low risk groups with median risk value as the boundary. Kaplan–Meier survival analysis showed that the survival rate of patients in the high-risk group was lower (*p* < 0.0001). (**c**) Receiver operating characteristic (ROC) curve was drawn to verify the accuracy of Cox survival analysis model, and the area under the curve for predicting 3-year survival rate was 0.784, indicating the high predictive power. (**d**) Heatmap of 6 hub mRNAs’ expression between high and low risk samples. (**e**) Correlation diagram of 6 hub mRNAs, red color represents positive correlation, blue color represents negative correlation, and the size of the circle reflects the *p*-value. (**f**) Expression of six hub mRNAs between AML and normal control samples.

**Figure 5 genes-11-00868-f005:**
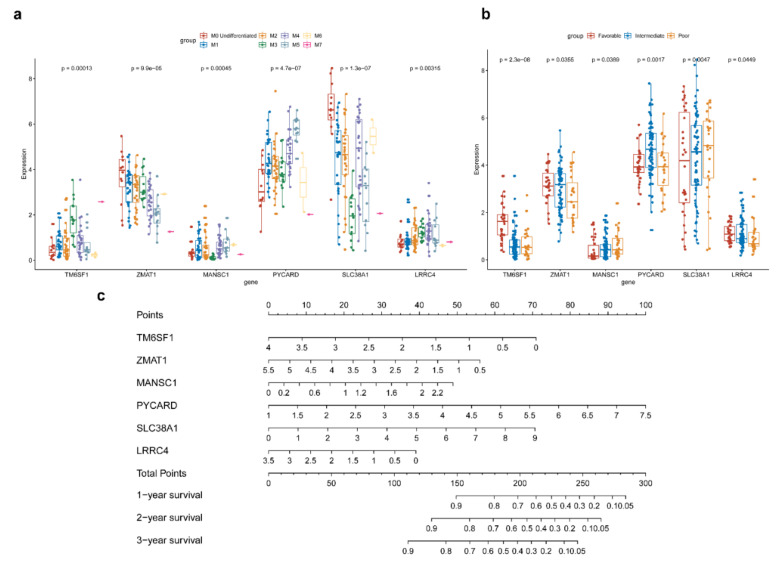
Identification of hub prognostic mRNAs’ expression difference between different AML subtypes and cytogenetic risk groups and construction of nomogram. (**a**) The expressions of the 6 mRNAs in different subtypes were significantly different, and the expression distribution trend are obvious differences among these mRNAs. (**b**) The distribution of 6 mRNAs in different cytogenetic risk groups was significantly different. (**c**) Nomogram was drawn to establish a more refined scoring system to evaluate the impact of hub mRNAs on prognosis.

**Figure 6 genes-11-00868-f006:**
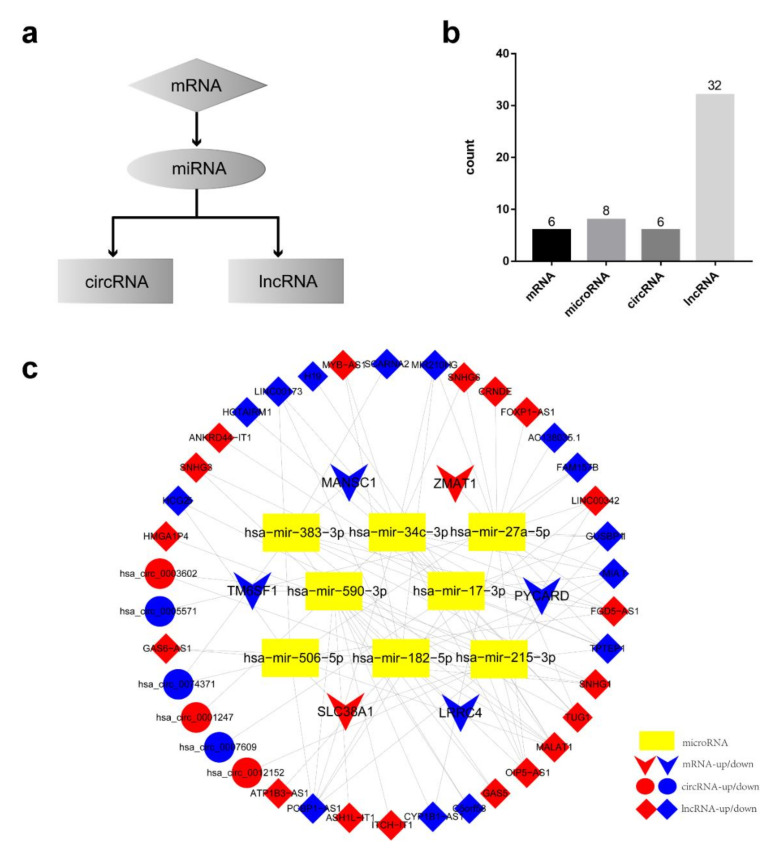
Establishment of AML prognostic competitive endogenous RNA (ceRNA) regulatory network. (**a**) ceRNA network construction process. Determined the miRNAs, long non-coding RNAs (lncRNAs), and circular RNAs (circRNAs) in ceRNA network based on 6 prognostic hub mRNAs. (**b**) The number of each RNAs in ceRNA network. There were 6 circRNAs, 32 lncRNAs, 8 miRNAs, and 6 mRNAs. (**c**) Interaction diagram of multiple endogenous RNAs in ceRNA network.

**Figure 7 genes-11-00868-f007:**
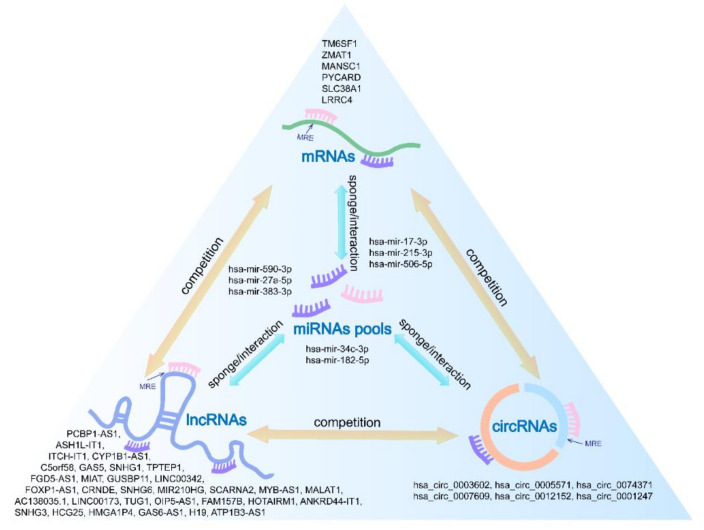
Molecular mechanism of the AML hub prognostic ceRNA network.

**Figure 8 genes-11-00868-f008:**
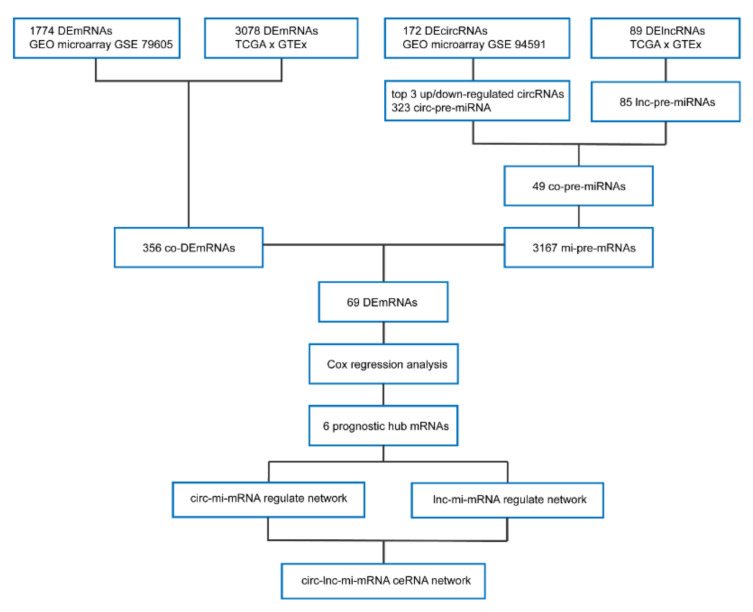
Analysis flow chart of this study.

**Table 1 genes-11-00868-t001:** The clinicopathological and molecular characteristics of AML patients.

	Alive (*n* = 54)	Dead (*n* = 97)	Total (*n* = 151)
**Gender**			
Female	24(44.4%)	44(45.4%)	68(45.0%)
Male	30(55.6%)	53(54.6%)	83 (55.0%)
**FAB Classification**			
M0	5(9.3%)	10(10.3%)	15(9.9%)
M1	11(20.4%)	24(24.7%)	35(23.2%)
M2	14(25.9%)	24(24.7%)	38(25.2%)
M3	11(20.4%)	4(4.1%)	15(9.9%)
M4	8(14.8%)	21(21.6%)	29(19.2%)
M5	5(9.3%)	10(10.3%)	15(9.9%)
M6		2(1.3%)	2(1.3%)
M7		1(0.7%)	1(0.7%)
Not Classified		1(0.7%)	1(0.7%)
**Cytogenetic Risk Group**			
Favorable	19(35.2%)	12(12.4%)	31(20.6%)
Intermediate	29(53.7%)	53(54.6%)	82(54.3%)
Poor	6(11.1%)	30(30.9%)	36(23.8%)
Missing data		2(2.1%)	2(1.3%)

**Table 2 genes-11-00868-t002:** Results of gene ontology (GO) enrichment analysis of DEmRNAs.

Ontology	Term	Description	Count	Adj. *p*-Value
BP	GO:0043312	neutrophil degranulation	56	5.38148E-26
BP	GO:0001819	positive regulation of cytokine production	30	9.621E-07
BP	GO:0032640	tumor necrosis factor production	15	0.000148136
BP	GO:0032757	positive regulation of interleukin-8 production	8	0.000698632
BP	GO:0032609	interferon-γ production	11	0.001628489
CC	GO:0034774	secretory granule lumen	33	1.18956E-13
CC	GO:0060205	cytoplasmic vesicle lumen	33	1.98605E-13
CC	GO:0101002	ficolin-1-rich granule	23	1.59718E-11
CC	GO:0005766	primary lysosome	14	2.12213E-05
CC	GO:0005802	trans-Golgi network	16	0.000131017
MF	GO:0030246	carbohydrate binding	17	0.007065521
MF	GO:0008329	signaling pattern recognition receptor activity	5	0.007065521
MF	GO:0038187	pattern recognition receptor activity	5	0.007065521

**Table 3 genes-11-00868-t003:** Results of KEGG enrichment analysis of DEmRNAs.

TermID	Description	Count	Adj. *p*-Value
hsa04640	Hematopoietic cell lineage	11	0.003548969
hsa04612	Antigen processing and presentation	9	0.008063516
hsa05150	Staphylococcus aureus infection	9	0.026866028
hsa04142	Lysosome	10	0.030064746
hsa04145	Phagosome	11	0.034084109
hsa05152	Tuberculosis	12	0.034084109
hsa05132	Salmonella infection	13	0.042405443

**Table 4 genes-11-00868-t004:** Results of KEGG enrichment analysis of DEmRNAs.

Gene	Univariate Cox		Multivariate Cox
	HR	*p*-Value	*p*-Value
ZMAT1	0.334297693	0.00048	0.0412
MANSC1	2.233815416	0.0021	0.0775
TM6SF1	0.490252653	0.00312	0.0443
PYCARD	3.194180152	0.00478	0.0081
RAB31	1.426030724	0.00971	
SLC38A1	1.630159352	0.01083	0.0036
IGF2R	1.466708157	0.01339	
CLEC7A	1.315460822	0.01363	
UBXN11	1.805081888	0.01387	
SLC8A1	1.587431529	0.01947	
NRIP1	1.521500351	0.01983	
LRRC4	0.449053605	0.02275	0.0812
SLC9A7	1.594442485	0.02472	
FGD4	1.34192418	0.04989	

**Table 5 genes-11-00868-t005:** Composition of AML prognosis-related ceRNA regulatory network.

Type	Hub RNA
mRNA	TM6SF1	ZMAT1	MANSC1	PYCARD	SLC38A1	LRRC4
miRNA	hsa-mir-590-3p	hsa-mir-27a-5p	hsa-mir-383-3p	hsa-mir-34c-3p		
	hsa-mir-215-3p	hsa-mir-506-5p	hsa-mir-182-5p	hsa-mir-17-3p		
circRNA	hsa_circ_0003602	hsa_circ_0005571	hsa_circ_0074371			
	hsa_circ_0007609	hsa_circ_0012152	hsa_circ_0001247			
lncRNA	PCBP1-AS1	ASH1L-IT1	ITCH-IT1	CYP1B1-AS1	C5orf58	GAS5
	SNHG1	TPTEP1	FGD5-AS1	MIAT	GUSBP11	LINC00342
	FOXP1-AS1	CRNDE	SNHG6	MIR210HG	SCARNA2	MYB-AS1
	MALAT1	AC138035.1	LINC00173	TUG1	OIP5-AS1	FAM157B
	HOTAIRM1	ANKRD44-IT1	SNHG3	HCG25	HMGA1P4	GAS6-AS1
	H19	ATP1B3-AS1

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
