# Peer review of "Identification of circRNA-lncRNA-miRNA-mRNA Competitive Endogenous RNA Network as Novel Prognostic Markers for Acute Myeloid Leukemia"

_genes, 2020, doi:10.3390/genes11080868_

Round 1
Reviewer 1 Report
In this manuscript Cheng et al. identified a potential ceRNA network comprising of circRNAs, lncRNAs, miRNAs and mRNAs in AML for the first time. This might help in understanding the post-transcriptional regulatory mechanism of AML. The results are interesting given the significance of the studied area. However, there are several major concerns that need to be addressed in order to support the conclusions mentioned:
i) As mentioned by the authors comparing RNA sequencing data from AML patients' bone marrow to normal peripheral blood samples is inconsistent and might not reveal the actual biology behind AML prognosis. There are transcriptome profiling studies as well as non-coding RNA profiling studies where AML BM samples have been compared to normal BM samples. It would be really beneficial if authors can use those datasets to strengthen their conclusions.
ii) For the transcriptome data analysis, readouts from RNA sequencing and microarray has been used in combination. As the chemistry behind these two techniques is quite different it is difficult and not appropriate to combine their outputs. Did authors use any specific normalization technique/correction methods to rule out the baisedness arising in this context?
iii) Acute myeloid leukemia (AML) is a heterogenous disease and is classified into several subtypes. Each subtype has to be treated in a different manner and has a different outlook which is in turn giving rise to the concept of personalized medicine these days. It would be interesting if authors provide their results in this aspect and try to find out how the ceRNA networks change with various AML subtypes. Do the biomarkers change with subtype and how do subtype specific biomarkers play a role in prognosis of the disease?
iv) Authors use 407 normal peripheral blood samples and 151 AML patients' bone marrow samples. The reason for this dataset imbalance i.e. using more normal samples as compared to the disease samples is not clear. Please comment.
v) The biomarkers and networks determined are based on bioinformatic predictions. As such adding some wet lab validation experiments (e.g. qPCR) would strengthen the existence of these biomarkers/networks within AML system.
vi) Authors have provided the role of the selected mRNAs, lncRNAs in AML. It has been explained individually how they are linked to AML. However a detailed molecular mechanism involving all the players i.e. circRNA, lncRNA, mRNAs and miRNAs and how all of them together are contributing to AML biology is not clear. It would be interesting if authors provide a combined molecular mechanism schematic showing the contribution of each selected player in AML.
vii) The clinical significance of this study is still not clear. How do authors think this work might contribute to future therapeutics? As because this study deals with a network of molecules contributing to the ceRNA concept do they think they would be able to modulate the expression of all them (i.e. circRNA, lncRNA, mRNA, miRNA) simultaneously to modulate AML prognosis?
viii) It is quite difficult to read the GO terms and KEGG pathways from the circos plot (Figure 1d-e). A simple bar diagram with the GO terms/KEGG pathway vs p-value will be a more clear representation.
ix) A conclusive network of 6 circRNAs, 32 lncRNAs, 8 miRNAs and 6 mRNAs was determined. It would be interesting if authors use a scoring method (e.g. ceRNA score) to determine the strength of each of these interactions and focus on the most strong interactions. This in turn will also help narrow down the network with high chances of obtaining relevant players contributing to AML.
Author Response
Dear Editors and Reviewers:
On behalf of my co-authors, we thank you very much for giving us an opportunity to revise our manuscript, we appreciate editor and reviewers very much for their positive and constructive comments and suggestions on our manuscript entitled “Identification of circRNA-lncRNA-miRNA-mRNA competitive endogenous RNA network as novel prognostic markers for acute myeloid leukemia”. (ID: genes-870262).
We have studied reviewer’s comments carefully. Those comments are all valuable and very helpful for revising and improving our paper, as well as the important guiding significance to our researches. We have tried our best to revise our manuscript according to the comments, which we hope meet with approval. Revised portion are marked using the track changes mode in MS Word and highlighted with different colors. The main corrections in the paper and the responds to the reviewer’s comments are as flowing:
Responses to the comments of Reviewer #1
First of all, we appreciate all your scientific and careful comments, we have benefited a lot. We tried our best to modify our manuscript according to your advice and make improvements to the entire article with your help. All the changes in this revised version is highlighted with yellow color.
Comments 1: As mentioned by the authors comparing RNA sequencing data from AML patients' bone marrow to normal peripheral blood samples is inconsistent and might not reveal the actual biology behind AML prognosis. There are transcriptome profiling studies as well as non-coding RNA profiling studies where AML BM samples have been compared to normal BM samples. It would be really beneficial if authors can use those datasets to strengthen their conclusions.
Response 1: Your questions and ideas are very helpful to improve the scientific nature of this article. Regarding the sample issue, we sincerely explain to you in detail:
â…°. First, the objective limitation is that there are only AML tumor samples (monocytes from the bone marrow) in the TCGA database, but no normal samples. In order to obtain a sufficient number of normal controls, we chose to obtain matched AML’s normal controls from the UCSC Xena database which is integrated with TCGA, GTEx, and many other databases. Unfortunately, the matched control for this database is not normal bone marrow mononuclear cells, but peripheral blood mononuclear cells. We combined this issue with multiple bioinformatics analysis articles focused on AML (Sun, Huang, Wang, Zhang, & Wang, 2018; Wang et al., 2019), and decided to use normal peripheral blood monocytes, which are also belonged to the blood system, as a normal control to TCGA tumor samples, and compared the differential expressed genes of these two monocytes.
ⅱ. Secondly, consistent with your comments, we also think that only analysis of the differential expressed mRNAs (DEmRNAs) between AML patients' bone marrow to normal peripheral blood samples are not rigorous enough. For this reason, we downloaded a microarray containing AML and Normal bone marrow monocytes from the GEO database, and identified the DEmRNAs between the different BM samples (as showed in Page 3, line 102-104,and Page 4, line 157-158, and Figure 1b). We believe that these DEmRNAs are in full compliance with the standard. However, considering the small sample size in this microarray (2 AML samples and 2 normal samples), we intersected GEO-DEmRNAs with TCGA X GTEx-DEmRNAs in order to improve the accuracy and credibility of this analysis. We have done our best to improve the reliability of the analysis by using multi-platform analysis, expanding the sample size, and obtaining common DEmRNAs. Thanks again for your help!
References:
Sun, X., Huang, S., Wang, X., Zhang, X., & Wang, X. (2018). CD300A promotes tumor progression by PECAM1, ADCY7 and AKT pathway in acute myeloid leukemia. Oncotarget, 9(44), 27574-27584. doi:10.18632/oncotarget.24164
Wang, J. D., Zhou, H. S., Tu, X. X., He, Y., Liu, Q. F., Liu, Q., & Long, Z. J. (2019). Prediction of competing endogenous RNA coexpression network as prognostic markers in AML. Aging (Albany NY), 11(10), 3333-3347. doi:10.18632/aging.101985
Comments 2: For the transcriptome data analysis, readouts from RNA sequencing and microarray has been used in combination. As the chemistry behind these two techniques is quite different it is difficult and not appropriate to combine their outputs. Did authors use any specific normalization technique/correction methods to rule out the biasedness arising in this context?
Response 2: We are very grateful for your professional questions and suggestions.
â…°. First, thank you for reminding us to clarify the normalization method. We used the "normalizeBetweenArrays" function of the "limma" R package for normalization in both sequencing and microarray data processing to ensure that the output results obtained are comparable. According to your suggestion, we have added it to the methods (Page 3, line 107-108) and result (Page 4, line 154-155) sections.
â…±. At the same time, for the gene expression counts obtained in the GTEx database, we performed log2 (FPKM+0.001) conversion according to the description on the GTEx official website. The gene expression in the TCGA database and some GEO microarray in needed were carried out the conversion of log2 (FPKM+1)
â…². In addition, in order to avoid possible misunderstandings due to our inaccurate expression, we would like to explain to you that we did not merge the expression profiles of TCGA x GTEx and GEO into one matrix, but made the analysis of differential expressed genes separately and take the intersection. This step is also described in (Page 4, line 158-159).
We value your opinion very much, so we have explained it in as much detail as possible, and hope to give you a satisfactory answer!
Comments 3: Acute myeloid leukemia (AML) is a heterogenous disease and is classified into several subtypes. Each subtype has to be treated in a different manner and has a different outlook which is in turn giving rise to the concept of personalized medicine these days. It would be interesting if authors provide their results in this aspect and try to find out how the ceRNA networks change with various AML subtypes. Do the biomarkers change with subtype and how do subtype specific biomarkers play a role in prognosis of the disease?
Response 3: Thank you very much and we strongly agree with your suggestion that AML Subtype should be considered heavily in the study. We have added the following work on this comment:
â…°. First, we added a statistical table describing the clinical information of 151 AML samples from the TCGA database (table 1, Page 6, Line 176), focusing on the proportion of M0-M7 subtypes and 3 cytogenetic risk group in patients with survival and death.
â…±. Secondly, because the prognostic ceRNA network in this study was established based on hub prognostic mRNAs, we added analysis of 6 hub mRNAs expression between M0-M7 subtypes according to your recommendations (Figue 5a, Page 14, line255 -266). The result showed that the expressions of the 6 mRNAs in different subtypes were significantly different, and the expression distribution trend are obvious differences among these mRNAs.
TM6SF has the highest expression value in M7 subtype and the lowest expression value in M6 (P=0.00013), ZMAT1 has the highest expression value in M0 subtype and the lowest expression value in M7 subtype (P<0.0001), MANSC1 was mostly expressed in M6 subtype and the minimally expressed in M3 subtype (P =0.00045), PYCARD has the highest expression value in M5 subtype, and the lowest expression value in M7 subtype (P<0.0001), SLC38A1 has the highest expression value in M0 subtype and the lowest expression value in M3 subtype (P<0.0001), LRRC4 has the highest expression value in M4 subtype and he lowest expression value in M6 subtype (P=0.00315).
â…². Through your inspiration, we also analyzed the differences in the expression levels of 6 hub mRNAs among the three cytogenetic risk groups (Figure 5b). It was also found that the distribution of 6 mRNAs in different cytogenetic risk groups was significantly different. TM6SF (P<0.0001) and LRRC4 (P=0.0449) had the highest expression value in the samples with the favorable karyotype, ZMAT1 (P=0.0355), MANSC1 (P=0.0389), and PYCARD (P=0.0017) has the highest expression value in the intermediate karyotype samples, while SLC38A1 (P=0.0047) has the highest expression in the poor outcome group samples. These results were consistent with the expression trend of hub mRNAs in AML compared to normal controls, and were also consistent with the results of multivariable COX analysis.
ⅳ. In order to demonstrate the role of the 6 core mRNAs further visually in the prognosis of AML, we added the nomogram to better show the relationship between their expression and prognosis(Figure 5c, Page 15, line276-278).
The above is our supplementary analysis work to answer this question. We hope to improve the scientificity of this research with your help. Thank you again for your helpful comments!
Comments 4: Authors use 407 normal peripheral blood samples and 151 AML patients' bone marrow samples. The reason for this dataset imbalance i.e. using more normal samples as compared to the disease samples is not clear. Please comment.
Response 4: We are very grateful for your kind reminder on this question. As we answered in comment 1, There is no normal control sample in the TCGA database, after referring to the grouping method of multiple latest bioinformatics articles on AML, we selected normal monocytes (407 samples) in GTEx database as normal control from the point of view of sample size. What's more, considering that we have no way to obtain the clinical information of the 407 normal samples, we are concerned that selecting a certain number of samples from total 407 normal samples cannot guarantee the randomness of the selection, but may bias the results, So we kept all 407 normal samples for differential expressed gene analysis.
Comments 5: The biomarkers and networks determined are based on bioinformatic predictions. As such adding some wet lab validation experiments (e.g. qPCR) would strengthen the existence of these biomarkers/networks within AML system.
Response 5: We quite agree with your suggestion, which will enrich the scientific conclusion of this study.
â…°. First, our research analyzed multiple databases and sought common differential genes for analysis, we also combined circRNA-miRNA, lncRNA-miRNA, and miRNA-mRNA multiple targeting relationship prediction, these fully guaranteed the screened ceRNAs with high reliability and close correlation with AML. We believe that the results of database analysis and targeted predictions together strengthen the presence of ceRNA in the AML system.
â…±. Secondly, univariate COX analysis, multivariate COX analysis, Kaplan-Meier survival analysis, and nomogram have been applied to the screening of hub prognostic mRNA. Our results also confirm that the survival rate of patients in the high-risk group is lower than that of the low risk group, the AUC of ROC curve was close to 0.8, which also reflects the high accuracy and credibility of this screening model. All of these ensure that the ceRNA network we constructed is highly correlated with the prognosis of AML and has strong predictive power.
ⅲ. Thirdly, most of the ceRNAs predicted in this study have been confirmed by other researchers or public databases that they are indeed significant in the progression and prognosis of AML and other tumors, which we have elaborate as fully as possible in the Discussion section(Page 20-21). These literatures are strong support for these result in our research, and also confirm the existence and role of ceRNA biomarkers in AML.
â…³. Finally, although we also believe that subsequent verification of expression, phenotype, function, and targeting relationship of the ceRNAs would be icing on the cake of our study, there are many practical problems in the verification process that cannot be overcome in a short period of time: the department we work at is an ophthalmology hospital, and it is more difficult to collect blood disease samples. We are willing to find partners to gradually complete the complete ceRNA verification experiment in future, and look forward to communicating with you. We hope that you will be satisfied with our answer and understand our current situation. Thanks again!
Comments 6: Authors have provided the role of the selected mRNAs, lncRNAs in AML. It has been explained individually how they are linked to AML. However, a detailed molecular mechanism involving all the players i.e. circRNA, lncRNA, mRNAs and miRNAs and how all of them together are contributing to AML biology is not clear. It would be interesting if authors provide a combined molecular mechanism schematic showing the contribution of each selected player in AML.
Response 6: We are very grateful for your suggestions. These ideas can help us better present the results of this research. lncRNAs, circRNAs, and protein-coding mRNAs all have MRE regions that can bind microRNAs (miRNAs), they compete for limited miRNAs and form a ceRNA regulatory network. When mRNA competes to bind miRNAs, its stability decreases, the translation process is blocked, and gene expression is affected, and the binding of miRNAs with circRNAs and lncRNAs can help mRNA escape this fate and complete the protein coding process. This is the molecular mechanism by which the non-coding RNA in the ceRNA network we have established regulates mRNA translation, thereby regulating the progression and prognosis of AML (as mentioned in Page 2, linen60-65). According to your suggestion, we have drawn a new mechanism diagram to show the principle of the ceRNA network more clearly and intuitively (Figure 7). At the same time, considering that the interaction diagram of the ceRNA network in the primary Figure 6 is relatively complicated, we have added one new Sankey diagram in order to do our best to better show our results (supplementary Figure 1).
Comments 7: The clinical significance of this study is still not clear. How do authors think this work might contribute to future therapeutics? As because this study deals with a network of molecules contributing to the ceRNA concept do they think they would be able to modulate the expression of all them (i.e. circRNA, lncRNA, mRNA, miRNA) simultaneously to modulate AML prognosis?
Response 7: Thank you for your question, we understand and agree with your consideration very much.
â…°. First, the prognosis of AML is difficult to predict, which brings great difficulties to clinical diagnosis and treatment. The purpose of this study is to find reliable markers for the prediction of AML prognosis through our accurate and rich data analysis and verification. Clinically, prognostic biomarkers can be used to guide patient management, instruct individualized adjuvant treatment for high-risk of patients at early stage and prevent the development of potentially untreatable stage. We have identified six hub mRNAs through multiple screenings, and believe that they can be used in clinical diagnosis. If these prognostic markers can help clinicians predict the patient's disease process and outcome, it will be the result we most hope for and certain contributions that we can made to promote clinical diagnosis and treatment of AML.
â…±. At the same time, as we all know, non-coding RNA is the post-transcriptional regulatory mechanism that has not been emphasized and understood in the past. In order to serve more basic and clinical research of AML, we have also constructed a ceRNA network that is highly correlated with these 6 hub mRNAs through multiple targeting predictions, hoping that the related ceRNA can be referenced as an important process of modulating mRNAs when researchers use these 6 mRNAs to judge the prognosis of AML to increase the understanding of the potential regulatory pathways of hub prognostic mRNAs. We also expect that with the in-depth development of non-coding RNA research, all compounds in ceRNA network can be easily modulated through regulating one of them, thereby realize the leap-forward development of effective treatment for AML.
â…². Therefore, ceRNA is a steady-state population that checks and balances each other. We also believe that regulating the expression of all components is complicated and difficult to achieve at present stage. But we only need to break the balance between the various components of ceRNA in the process of promoting the development of AML, and there is hope to realize the innovation of treatment methods. Just as it is impossible for us to use inhibitors or agonists of all genes of the pathogenic pathway to treat diseases, but we can regulate a core gene of this pathway to achieve better curative effects. We do our best to think about and answer your questions, and enjoy the process of exchanging ideas with you, hope these answers will satisfy you, thank you again!
Comments 8: It is quite difficult to read the GO terms and KEGG pathways from the circos plot (Figure 1d-e). A simple bar diagram with the GO terms/KEGG pathway vs p-value will be a more clear representation.
Response 8: We are sorry that the images provided do not show the experimental results intuitively. We have modified the GO and KEGG enrichment results in Figure 1 (Figure 1c-d in the revision version), and replaced it from the circle graph to a clearer and more concise bar diagram. Thank you for such kind reminders and suggestions, we will pay attention to it in future picture editing.
Comments 9: A conclusive network of 6 circRNAs, 32 lncRNAs, 8 miRNAs and 6 mRNAs was determined. It would be interesting if authors use a scoring method (e.g. ceRNA score) to determine the strength of each of these interactions and focus on the most strong interactions. This in turn will also help narrow down the network with high chances of obtaining relevant players contributing to AML.
Response 9: Your suggestions are particularly inspiring and provide a so good idea for more precise prognostic markers of AML, which is what we want to achieve in the research process! In order to establish a more refined scoring system to evaluate the impact of core mRNA on prognosis, we have added the respective risk factors for the 6 hub prognostic mRNAs (Page 14, line 244-247), and added a nomogram (Figure 5c). We are also very glad to score the strength of the interaction between ceRNAs according to your suggestions, but due to the limitation that the expression data of various RNAs are obtained from different platforms and different samples, we currently failed to merge them together for co-expression analysis or other analysis for calculation of ceRNAs’ correlation and connectivity. This is also one of the limitations of this research. We thank you very much for your reminder and have added it in the “Discussion” section (Page 20, line 451-455). We also hope to remind future readers of this article that if they expect to screen out the RNAs with the strongest correlation and interaction in the ceRNA network, they should consider the source of the sample when selecting data. Thank you again for your careful review and help of our manuscript!
Special thanks to you for your good comments.
We tried our best to improve the manuscript and made some changes in the manuscript. We would like to express our great appreciation for Editors/Reviewers’ valuable and kind work and wish you all the best with the covid-19 epidemic. Once again, thank you very much for your comments and suggestions.
Thank you and best regards!
Sincerely,
Zhi-Chong Wang, M.D, Professor
State Key Laboratory of Ophthalmology
Zhongshan Ophthalmic Center, Sun Yat-sen University
54 Xian Lie Nan Road
Guangzhou 510060, China
Phone: 86-020-8733-0379
Email: wangzhichong@gzzoc.com

Reviewer 2 Report
The paper “Identification of circRNA-lncRNA-miRNA-mRNA competitive endogenous RNA network as novel prognostic markers for acute myeloid leukemia” by Cheng and coworkers is submitted for consideration for publication in the journal Genes.
The authors obtained transcriptome data of AML and normal samples from available databases, and identified differentially expressed (DE) mRNAs, lncRNAs, and circRNAs. They identified prognostic 6 hub mRNAs through Cox regression model and divided the AML samples into high and low risk groups according to the risk score obtained by multivariate Cox regression. Survival analysis verified that the survival rate of the high-risk group was significantly reduced.
The study has some interesting findings and novelty, and the results are deserved a publication. However, I have some suggestions for improvement which I think will strengthen the significance and the presentation of the data.
The most important thing is that the figures and graphical presentation, both figures and table; should be significantly improved. For Figure 1 and b the volcano plots are not adding anything to the hierarchical clustering visualization and should be removed. Figure 1 c, the Wenn diagrams are of less importance and could be removed. Figure 1 d and e the legends are impossible to read and make no sense. Restructure to larger versions or remove. Table 1; it seems like the GO-terms are representing much of the same counts; it seems not necessary to present all this data with almost similar terms in one table. Figure 2b; impossible to read; just be presented in larger scale, or in supplementary. Figure 2d; what is the criteria of naming some Cic-RNA (i.e. two reads, and five greens)? Figure 2f, impossible to read terms, seem not relevant.
Figure 3f should be better presented as box plot rather than dot plots.
In general, the discussion should be improved. It seems to be more of a review character, rather than discussion of the actual findings in the study.
Author Response
Dear Editors and Reviewers:
On behalf of my co-authors, we thank you very much for giving us an opportunity to revise our manuscript, we appreciate editor and reviewers very much for their positive and constructive comments and suggestions on our manuscript entitled “Identification of circRNA-lncRNA-miRNA-mRNA competitive endogenous RNA network as novel prognostic markers for acute myeloid leukemia”. (ID: genes-870262).
We have studied reviewer’s comments carefully. Those comments are all valuable and very helpful for revising and improving our paper, as well as the important guiding significance to our researches. We have tried our best to revise our manuscript according to the comments, which we hope meet with approval. Revised portion are marked using the track changes mode in MS Word and highlighted with different colors. The main corrections in the paper and the responds to the reviewer’s comments are as flowing:
Responses to the comments of Reviewer #2
We are very grateful for your recognition and encouragement of this research, which gave us great motivation and confidence. Your patient guidance also made us very rewarding. We highly approve of your Suggestions and have made modification one by one. We hope our work can satisfy you! Thanks again! All the changes in this revised version is highlighted with green color.
Comments 1: For Figure 1 and b the volcano plots are not adding anything to the hierarchical clustering visualization and should be removed. Figure 1c, the Wenn diagrams are of less importance and could be removed. Figure 1 d and e the legends are impossible to read and make no sense. Restructure to larger versions or remove.
Response 1: We are sorry that we have many shortcomings in the picture composition and construction. Thank you very much for your helpful suggestions to help us better present the research results. For Figure 1, we made the following revisions: â…°. we reconstructed the overall picture of Figure 1, deleted the volcano plot and Venn diagram. â…±. At the same time, considering the limitation that the legend of GO and KEGG results is not clear and the displayed terms are less in circle diagram format, we replaced it with a simple bar diagram to improve the readability of the picture.
Comments 2: Table 1; it seems like the GO-terms are representing much of the same counts; it seems not necessary to present all this data with almost similar terms in one table.
Response 2: Thank you for your help. Our primary table only showed the first 5 terms with the smallest adjust P value, which led to the display of many similar terms. We have carefully considered your suggestions and thought that the primary methods weakened the meaning of this table. In order to show more and relatively complete results of GO enrichment, we re-selected different terms and drew a brand new table2 (corresponding to the original table1). We hope our work can satisfy you.
Comments 3: Figure 2b; impossible to read; just be presented in larger scale, or in supplementary. Figure 2d; what is the criteria of naming some Cic-RNA (i.e. two reads, and five greens)? Figure 2f, impossible to read terms, seem not relevant.
Response 3: Thank you for your careful guidance. In response to the shortcoming in Figure 2, we have made the following changes:
â…°. Figure 2b in the first draft is to show the targeting effect of lncRNA-miRNA. Due to our negligence in the image layout, it was compressed to a too small size which affected its readability. Therefore, we split Figure 2 into Figure 2 and Figure 3, and provide a larger size of this targeting relationship picture in Figure 2c.
â…±. In response to the circRNA naming problem you raised for Figure 2d in the first draft, the international universal naming method for circRNAs is has_circ_ seven Arabic numerals, what we show in the volcano plot is the microarray’s annotations. We forgot to transform them into the official name due to our negligence and we feel very sorry about that. Considering the layout of the entire picture and the importance of each sub-picture when regrouping the picture, we decided to remove this volcano diagram.
â…². Figure 2f in the first draft shows the targeting relationship of circRNA-miRNA, which has been reorganized into Figure 3e and provided a larger size for clear display. Thanks again for your kind comments!
Comments 4: Figure 3f should be better presented as box plot rather than dot plots.
Response 4: Thank you for sharing with us the tips for selecting picture types. We have converted dot plots to box plots in Figure 4f in the revised draft.
Comments 5: In general, the discussion should be improved. It seems to be more of a review character, rather than discussion of the actual findings in the study.
Response 5: We thank you very much for pointing out this issue, which helped us think about our research results and improve the discussion section. We studied and discussed this issue in detail, and did our best to modify the overall framework and details of the discussion. We used our summary and discussion of the results of this research to replace too many statements of the research results of others and look forward to satisfying you!
We appreciate for your academic guidance to help us make the manuscript better!
We tried our best to improve the manuscript and made some changes in the manuscript. We would like to express our great appreciation for Editors/Reviewers’ valuable and kind work and wish you all the best with the covid-19 epidemic. Once again, thank you very much for your comments and suggestions.
Thank you and best regards!
Sincerely,
Zhi-Chong Wang, M.D, Professor
State Key Laboratory of Ophthalmology
Zhongshan Ophthalmic Center, Sun Yat-sen University
54 Xian Lie Nan Road
Guangzhou 510060, China
Phone: 86-020-8733-0379
Email: wangzhichong@gzzoc.com

Round 2
Reviewer 1 Report
The authors have addressed all my concerns satisfactorily.
Reviewer 2 Report
My main concerns have been addressed, and the manuscript has significantly improved. Some minor spell checks and number checks are just needed (I.e. Table 1 Female not Famale and the numbers of percent did not match).